# Biological Prognostic Value of miR-155 for Survival Outcome in Head and Neck Squamous Cell Carcinomas: Systematic Review, Meta-Analysis and Trial Sequential Analysis

**DOI:** 10.3390/biology11050651

**Published:** 2022-04-24

**Authors:** Mario Dioguardi, Francesca Spirito, Diego Sovereto, Lucia La Femina, Alessandra Campobasso, Angela Pia Cazzolla, Michele Di Cosola, Khrystyna Zhurakivska, Stefania Cantore, Andrea Ballini, Lorenzo Lo Muzio, Giuseppe Troiano

**Affiliations:** 1Department of Clinical and Experimental Medicine, University of Foggia, Via Rovelli 50, 71122 Foggia, Italy; spirito.francesca97@gmail.com (F.S.); diego_sovereto.546709@unifg.it (D.S.); lucia_lafemina.560051@unifg.it (L.L.F.); alessandra.campobasso@unifg.it (A.C.); elicio@inwind.it (A.P.C.); dott.dicosola@gmail.com (M.D.C.); khrystyna.zhurakivska@unifg.it (K.Z.); lorenzo.lomuzio@unifg.it (L.L.M.); giuseppe.troiano@unifg.it (G.T.); 2Department of Basic Medical Sciences, Neurosciences and Sensory Organs, University of Bari “Aldo Moro”, 70124 Bari, Italy; stefaniacantore@pec.omceo.bari.it (S.C.); andrea.ballini@me.com (A.B.); 3Faculty of Dentistry (Fakulteti i Mjekësisë Dentare-FMD), University of Medicine, 1001 Tirana, Albania; 4Department of Precision Medicine, University of Campania “Luigi Vanvitelli”, 80138 Naples, Italy

**Keywords:** miR-155, microRNA, OSCC, HNSCC, non-coding RNA, oral cancer

## Abstract

**Simple Summary:**

Head and neck squamous cell carcinoma (HNSCC) is the neoplasm with the highest incidence in the head and neck regions. More than 350,000 new cases of carcinoma are diagnosed worldwide every year, but the prognosis has not improved significantly in the past few decades. In addition to smoking and alcohol habits, which represent the main risk factors for HNSCC, deregulation of non-coding RNAs have been identified as negative prognostic survival factors. The expression of miR-155 appears to be altered in many neoplasms and its different expression in tumor tissues can represent a prognostic biomarker of survival. By carrying out a systematic review and meta-analysis of the data in the current international literature, we aim to provide the most up-to-date data on the different expression miR-155 and to correlate those data with the prognostic indices of survival in HNSCC.

**Abstract:**

Head and neck squamous cell carcinoma (HNSCC) is one of the most common cancers worldwide; in fact, it is among the top six neoplasms, with an incidence of about 370,000 new cases per year. The 5-year survival rate, despite chemotherapy, radiotherapy, and surgery for stages 3 and 4 of the disease, is low. MicroRNAs (miRNAs) are a large group of small single-stranded non-coding endogenous RNAs, approximately 18–25 nucleotides in length, that play a significant role in the post-transcriptional regulation of genes. Recent studies investigated the tissue expression of miR-155 as a prognostic biomarker of survival in HNSCC. The purpose of this systematic review is, therefore, to investigate and summarize the current findings in the literature concerning the potential prognostic expression of tissue miR-155 in patients with HNSCC. The revision was performed according to PRISMA indications: three databases (PubMed, Scopus, and the Cochrane Register) were consulted through the use of keywords relevant to the revision topic. Totally, eight studies were included and meta-analyzed. The main results report for the aggregate HR values of 1.40 for OS, 1.36 for DFS, and 1.09 for DPS. Finally, a trial sequencing analysis was also conducted to test the robustness of the proposed meta-analysis.

## 1. Introduction

Head and neck squamous cell carcinoma (HNSCC) is an aggressive life-threatening disease associated with high mortality rates [1]. The 5-year survival rate, despite chemotherapy, radiotherapy, and surgery for stages 3 and 4 of the disease, is low (only 30% of patients survive) [2].

HNSCCs are neoplasms that, depending on the epithelium of origin, are recognized as laryngeal squamous cell carcinoma (LSCC), oropharyngeal squamous cell carcinoma (OPSCC), hypopharyngeal squamous cell carcinoma (HSCC), and oral squamous cell carcinoma (OSCC). The latter is the carcinoma with the highest incidence in the head and neck region, corresponding to around 60% of HNSCCs [3].

The main risk factors are represented by smoking and alcohol consumption, with an important correlation for LSCC with the papilloma virus (HPV 16, HPV 18). HNSCC therapy consists of a surgical approach associated with chemotherapy and radiotherapy for HPV positive tumors; relapses occur despite treatments in 50% of the cases [4].

At the base of the etiopathogenetic processes of cancerization we have genetic alterations that involve mutations of the main oncogenes (RAS [5], RAF [6], GSP [7], JUN [8], FAS [9], ERBA [10], ABL [11], SIS [12], ERB2 [13] and FMS [14]) and oncosuppressor genes (APC [15], P53 [16], NF1 [17], VHL [18], RB [19], BCL2 [20], BRCA2 [21], PTCH [22], CD95 [23], ST5 [24], SWI/SNF [25], P16 [26], YPEL3 [27], ST7 [28] and ST14 [29]). Understanding of the genomic aberrations is linked with non-coding genes (such as microRNAs), and their effects on HNSCCs are still relatively limited [30,31,32,33,34,35]. Current evidence suggests that deregulation of microRNA genes plays an important role in HNSCC [30,31,32,33,34,35,36,37,38,39,40].

MicroRNAs (miRNA, miR) are a large group of small single-stranded non-coding endogenous RNAs, approximately 18–25 nucleotides in length, that play a significant role in the post-transcriptional regulation of genes through the interaction with 3′UTR of target mRNA [41]. The degradation or inhibition of their translation also plays an important role in cell development in differentiation, metabolism, proliferation, migration, induction of angiogenesis, and apoptosis; microRNAs are stable molecules that can be found not only in tissues but also in body fluids, such as blood, saliva, and urine [42].

Alterations in the expression of miR can manifest themselves in changes that lead to their upregulation or downregulation. The main upregulated miRs associated with HNSCC are miR-21, miR-27a (-3p), miR-31, miR-93, miR-134, miR-146, miR-155, miR-196a, miR-196b, miR-211, miR-218, miR-222, miR-372, and miR-373, while the main downregulators are let-7a, let-7b, let-7c, let-7d, let-7e, let-7f, let-7g, let-7i, miR-26a, miR-99a-5p, miR-137, miR-139-5p, miR-143-3p, miR-184, and miR-375 [43].

The changes in the expression of miR from countless studies have been linked to the survival prognosis of patients suffering from the neoformation.

There is considerable evidence linking the altered profile of miRs with well-defined disease states affecting the onset, progression, and metastasis of HNSCCs. Their possible use as potential prognostic biomarkers can help to identify potential clinical characteristics of malignancy, such as relapses, invasiveness, and metastases, in order to be able to modulate more effective surgical, chemotherapy, and radiotherapy treatments in relation to the prognosis of the patient. There also seems to be an association that links patients with HNSCC HPV + with an altered expression of miRs; in fact, in a study conducted by Emmett et al. in 2021 [44], which compared miRs stratified by HPV status in HNSCCs, showed the presence of five miRs (miR-16-3p, miR-29a, miR-29c, miR-150, and miR-363) with different expressions. On the other hand, considering smoking patients, a very recent study on HNSCC has identified NNK (tobacco-specific nitrosamine 4-(methylnitrosamino)-1-(3-pyridyl)-1-butanone) present in cigarette smoke as a significant induced factor upregulation of miR-21 and miR-155 and downregulation of miR-422 [45].

As reported in the current scientific literature, the main miR investigated as a potential prognostic biomarker of survival for HNSCC is miR-21, together with other miRs [46], possibly including miR-31 and miR-155, both of which have been studied in recent years as potential biomarkers [47].

The miR-155 is located on chromosome 21 (21q21.3) in a region called the B lymphocyte integration cluster; its sequence is “UUAAUGCUAAUCGUGAUAGGGGU” [48].

Like most miRs, the expression of miR-155 is associated with the regulation of cell survival, growth, and chemosensitivity. It regulates the transcription of some genes, such as TP53INP1, ARID2, BACH1, and HIF, and it was initially implicated in the oncogenesis of hematopoietic tumors and subsequently in solid tumors of the breast, liver, lung, pancreas, thyroid, cervical tumors, and HNSCCs [49].

As previously stated, recent studies investigated the tissue expression of miR-155 as a prognostic biomarker of survival. In fact, Shi et al. [50] identified it as a potential prognostic biomarker for OSCC, which was confirmed by Wu et al. This identifies miR-155 as promoting oral cancer progression [48]. However, to date there are no systematic reviews on the prognostic role of miR-155 for HNSCCs.

Taking into account these premises, the aim of this systematic review and meta-analysis was to investigate the prognostic potential of miR-155 as a survival biomarker for HNSCCs, in light of new miR-155 studies.

## 2. Materials and Methods

### 2.1. Protocol

The drafting of the review was carried out following the indications of PRISMA (the Preferred Reporting Items for Systematic Reviews and Meta-Analysis) [51], which was the protocol with which the systematic review was performed and which was established before proceeding with the search and screening of records in data banks. The protocol was in fact registered in advance on INPLASY (the International Platform of Registered Systematic Review and Meta-Analysis Protocols), with registration number INPLASY202210119 and DOI number 10.37766/inplasy2022.1.0119.

### 2.2. Eligibility Criteria

The search for sources was directed toward all retrospective, prospective, and randomized trials that investigated the role of miR-155 in tumor tissues and HNSCCs; there also had to be a clear correlation with prognostic survival indices, including OS (overall survival), DFS (disease free survival), PFS (Progression free survival), RFS (relapse free survival), CSS (cancer specific survival), and the expression of miR-155.

The formulated PICO question was as follows: Is there a difference in the prognostic indices of survival between HNSCC patients with high tissue miR-155 expression versus those with low expression? The different points investigated were: (P)articipants (patients with HNSCC), (I)ntervention (altered expression of miR-155 in HNSCC), (C)ontrol (patients with HNSCC who have a low expression of miR-155), and (O)utcome (the difference in survival prognosis between patients with low and high miR-155 expression in HNSCC).

The exclusion criteria were the following: all studies with abstracts not in English, with no clear English translation available; studies that did not report data on the expression of miR-155 in tumor tissues; studies that did not report HNSCC prognostic survival data in relation to miR-155; systematic reviews and revisions (taken only into consideration as bibliographic sources); case reports; and case series.

Therefore, among the potentially eligible articles, it was decided to include those studies that, having investigated tissue miR-155 in relation to prognostic survival indices (OS, DFS, PFS, RFS, and CSS) for HNSCCs, reported the Hazard Ratio (HR), or Cox regression, or the Kaplan Meier survival curves.

### 2.3. Sources of Information, Research and Selection

For the research of the studies, the sources of information involved three reviewers (M.D., D.S., S.C.), who independently searched and subsequently selected the potentially eligible records in the electronic databases. The identified records are reported in different tables with the relative keywords and compared.

The search was performed on three different databases: the Scopus PubMed database, the Cochrane database, and the grey literature on the Open Grey database; Google Scholar was also consulted for sources not otherwise identifiable, and systematic reviews on the miR-155 were investigated in searches of further records. Therefore, the search and selection procedure predicted the following steps, decided in advance by the three reviewers: identification of the keywords to be used, identification of the databases, and independent research into the records, with the data reported in two different tables. Duplicate results were removed using the EndNote 9 software; the overlaps of studies that could not be uploaded to EndNote were manually removed after the screening phase, which involved the screening of potentially suitable articles (through the analysis of the title and the abstract) and the choice of articles to be included in the meta-analysis. The search for the records was completed on 1 February 2022 and a last update on the search was carried out on 14 February 2022. The keywords used for the databases were miR-155 and HNSCC; microRna and HNSCC; and prognosis, miR-155 and OSCC, LSCC, and miR-155. In addition, an update of the results was carried out on 12 April 2022, with the implementation of the Web of Science, Science Direct, and EBSCO databases.

### 2.4. Data Collection Process, Data Characteristics

The data to be extracted from the included articles were decided in advance by the three reviewers and were concerned with the lead author of the study, the date of publication, the country where the research was conducted, the type of squamous cell carcinoma involved, the number of patients, the miRs investigated, the value or type of cut-off between low and high expression for miR-155, and the HR values for the various prognostic survival indices.

Moreover, if only the Kaplan Meier survival curves were present, the Hazard Ratio was calculated using the Tierney method by extrapolating the data from the curve with Engauge Digitizer 4.1, and reported in a special Excel spreadsheet available online as Appendix A to the publication of Tierney et al. [52].

### 2.5. Risk of Bias in Individual Studies, Summary Measures, Summary of Results, Risk of Bias between Studies, Additional Measures

The risk of bias in the individual studies was assessed by an author (M.D.) using a tool for the assessment parameters derived from the Reporting Recommendations for prognostic studies of markers (REMARK) and the Quality Assessment Tool developed by the National Heart, Lung, and Blood Institute (NHLBI), (USA), for observational and cross-sectional purposes. The studies with a high risk of bias were considered for exclusion from the meta-analysis [53,54].

The evaluation of the heterogeneity was performed via the Higgins index (*I*^2^), and the Chi^2^, values of *I*^2^ higher than 75% led to a moderate heterogeneity of the data in the studies. The heterogeneity was also evaluated graphically through the analysis of the overlapping of the confidence intervals in the forest plot, through the graphical analysis of the funnel plot to search for any sources of heterogeneity on the presence of a publication bias.

The risk of bias between the studies was assessed graphically through the analysis of the overlaps of the confidence intervals, through the *I*^2^ inconsistency index (an *I*^2^ value greater than 75% was considered high and an analysis of the random effects), and through funnel plots.

The possibility of performing a sensitivity analysis was also evaluated in order to identify and exclude the source of heterogeneity, to investigate its effects on the pooled HR.

A subgroup analysis was also be performed with different meta-analyses as a function of the various prognostic indices of survival and as a function of the histological subtypes of HNSCC.

For the meta-analysis, and in particular for the calculation of the pooled HR, the software Reviewer Manager 5.4 (Cochrane Collaboration, Copenhagen, Denmark) was used. In particular, the GRADE pro-Guideline Development Tool online software (GRADEpro GDT, Evidence Prime, Hamilton, ON, USA) was used to assess the quality of the evidence [55]. The trial sequency analysis (TSA) was performed using Stata 13 (StataCorp, College Station, TX, USA) with the implementation of the R 4.2 software and by installing the idbounds and metacumbounds commands.

## 3. Results

### 3.1. Selection of Studies

The selection of the studies to be included started from several of the 765 bibliographic references identified by the three databases (Scopus, PubMed, and the Cochrane Central Register of Controlled Trials). Following the removal of duplicates, numerous references for the 549 records were obtained. After the analysis of the “abstract”, 21 potentially eligible articles were found. In addition, eight studies that fully met the inclusion and exclusion criteria were included. The keywords used for the research are indicated in Appendix A; the whole procedure of identification, selection, and inclusion of the studies is indicated on the flow chart in Figure 1.

### 3.2. Data Characteristics

Articles included in the meta-analysis were reported as follows: Jakob et al. (2019) [47], Hess et al. (2017) [56], Zhao et al. (2018) [57], Baba et al. (2016) [58], Shi et al. (2015) [50], Kim et al. (2018) [59], Bersani et al. (2018) [60], and Wu et al. (2020) [48].

The extracted data described in the Materials and Methods section were included in Table 1; if the survival data were expressed in the form of a Kaplan Meier curve, the HR was extracted according to the Tierney method.

The total number of patients in the included studies was 709, including 279 presenting with OSCC and 120 presenting with LSCC.

Only one study reported information on the histological grading of HNSCC, while the stratification of patients according to staging was clearly present in six of eight studies. The total of stage I and II patients was 125 out of 403, while there were 276 in stages III and IV. In terms of gender, 538 patients were male and 171 patients were female patients, in line with the most recent epidemiological studies [1].

Retrospective studies that reported prognostic data in association with miR-155 expression were published from 2015 to 2020, with mean follow-up periods ranging from 24 months (Baba et al. [58]) to 80 months (Kim et al. [59]). The studies that investigated OSCC only were the following five: Jakob et al. (2019) [47], Baba et al. (2016) [58], Shi et al. (2015) [50], Kim et al. (2018) [59], and Wu et al. (2020) [48].

Six studies reported the HR on OS as prognostic data, three studies reported the HR on DFS, and two studies reported the HR on PFS, while RFS was reported only by Jakob et al. (2019) [47] (Table 2). In three studies, the HR values were extrapolated from the Kaplan Meier curves and the values were reported (Table 2) in the meta-analysis software (Rev manager 5.4).

### 3.3. Risk of Bias in Studies

The risk of bias was assessed through parameters derived from REMARK. On the basis of the REMARK guidelines, a score from 0 to 3 was considered for each factor (Table 2). A risk of bias assessment was also conducted within the studies through the Quality Assessment Tool developed by the NHLBI for observational and cross-sectional purposes. The results were included in Appendix A.

### 3.4. Meta-Analysis

The meta-analysis was performed using the Rev manager 5.4 software. Three meta-analyses in total were carried out on the three main prognostic indices OS, PFS, and DFS, while for the RFS only one set of HR data was reported from the Jakob study (2019); for CSS, there were no HR data available in the included studies [47].

For the OS hazard ratio between high and low miR-155 expression, data from six studies were taken into consideration, and a fixed effects model was applied, as the heterogeneity between studies was low (*I*^2^ 28%).

The results of the first meta-analysis reported aggregate HR for OS. Between high and low miR-155 expression of 1.40, with the relative intervals of confidence [1.13 1.75]; heterogeneity was evaluated through Chi^2^ = 6.90 df = 5 (*p* = 0.23) and the Higgins index *I*^2^ = 28%; testing for the overall effect was Z = 3.01 (*p* = 0.003). The forest plot presents the black diamond in a position of worsening OS in relation to the high miR-155 expression (Figure 2).

For the second meta-analysis, the data taken into consideration concerned the HR on DFS with three studies included; there was an absence of heterogeneity between the data and a fixed effect model was applied.

Aggregate HR for DFS: Between high and low miR-155 expression of 1.36 with intervals of confidence [0.65 2.83], heterogeneity was evaluated through Chi^2^ = 0.02 df = 2 (*p* = 0.99) and the Higgins index *I*^2^ = 0; testing for the overall effect was Z = 0.81 (*p* = 0.42). The final result is that, even if the HR data were in favor of a worsening of the PFS, the central rhombus intersects the central line of the non-effect as well as all three single studies; the data were therefore devoid of statistical significance (*p* = 0.42) (Figure 3).

The third meta-analysis was conducted on PFS with only two studies included. Aggregate HR for PFS: Between high and low miR-155 expression of 1.09 with intervals of Confidence [0.53 5.15], heterogeneity was evaluated through Chi^2^ = 2.7 df = 1 (*p* = 0.10) and the Higgins index *I*^2^ = 63% and testing for the overall effect was Z = 0.23 (*p* = 0.82). The final result of HR for PFS showed no difference between high expression and low expression of miR-155; the rhombus was placed in the center of the no-effect line, with the two studies reporting opposite HR results (Figure 4).

### 3.5. Risk of Bias across Study, Sensitivity Analysis, Subgroup Analysis, Publication Bias

The risk of bias between the studies is considered to be low (*I*^2^ = 28%) and through the visual and graphic analysis of the confidence intervals, it does not emerge on the first meta-analysis (the only one of the three that reports statistically significant data with *p* = 0.02) sources of heterogeneity. Instead, the analysis of the funnel plot highlights how the study by Shi et al. (2015) could be a source of heterogeneous data with a possibility of risk of bias between studies (Figure 5) [50]. An evaluation of the publication bias was also conducted through the graphical analysis of the symmetry of the funnel plot, which appeared to be asymmetric. The presence of asymmetry demonstrated the potential possibility of having a bias of publication. In order to minimize the bias, a search was carried out for unpublished material in the gray literature, which also includes the presence of conference papers and materials that may not have been published due to the lack of significance of the results.

In fact, carrying out a sensitivity analysis and selectively excluding each single study from the analysis, it appears that excluding Zhao et al. (2018) or Baba et al. (2016) may represent a source of heterogeneity. The exclusion of one of these two studies caused the Higgins index to drop to 0% (Figure 6) [57,58].

A subgroup analysis was also conducted by dividing the OSCC studies from the HNSCC studies (no-OSCC). In the first subgroup (OSSC), the analysis was clearly in favor of a worsening of survival for patients with high expression of miR-155 in tissue cancer, while for the second subgroup, which concerned all non-oral squamous cell carcinomas of the head and neck, the data were slightly in favor of worsening.

The two subgroups, taken individually, were fairly homogeneous in *I*^2^ = 0% and *I*^2^ = 34%; the difference between the two subgroups was high, with Chi^2^ = 3.92, *I*^2^ = 74.5%, The risk of bias between the studies could provide a reason for the different locations of squamous cell carcinoma (localization of squamous cell carcinomas in the second subgroup: oropharynx *n* = 78, hypopharynx *n* = 71, larynx *n* = 120) (Figure 7).

### 3.6. Trial Sequential Analysis, Grade

Trial Sequential Analysis (TSA), was performed to evaluate the potency of the result of the first meta-analysis, adjusting the results to avoid type I and II errors. The program used was Stata 13 (StataCorp, College Station, TX, USA) with the integration of the R 4.2 software through the Metacumbounds commands as described by Miladinovic et al. [9]. The O’Brien–Fleminge spending function was used by applying random effects. The AIS (accrued information size) and subsequently APIS (a priori information size) commands were used via the Dialog BOX to determine the optimal sample size and the power of the results, assuming an RRR (reduction risk relative) of 38% and an alpha value equal to 5% (type 1 error) and beta at 20% (type 2 error) (Figure 8).

The TSA curve crossed the line Z = 1.98, and the crossing of the monitoring boundary before reaching the information size provided firm evidence of effect. The APIS graph showed that for an RRR of 38%, alpha 5%, and a power of 80%, the number of optimal patients is 475.

The authors, through the GRADE Guideline Development Tool (GRADEpro GDT), assessed the quality of the first meta-analysis result (Table 3). The results suggested that the quality of the evidence was low.

## 4. Discussion

To our knowledge, the present systematic review is currently the first review on miR-155 with meta-analysis conducted for HNSCC, in addition to trial sequential analysis, to evaluate the potency of the meta-analytic results. The review work included all prognostic indices as the subjects of investigation, but only for OS, DSF, and DPS was it possible to conduct a meta-analysis of the data. The included studies were eight in number, and the total number of patients was 709.

The only previous systematic review of the literature on the prognostic value of several miRs, including miR-155 in head and neck cancers, included miR-155 in a subgroup analysis that included only two studies (Shi et al. [48] and Hess et al. [56]). The results of this meta-analysis, HR of 1.866 (95% CI 1.047–3.326), were in line with our systematic review, which identified as many as six studies involving HR on OS [61].

The prognostic value of the altered expression was also investigated for other tumors. In fact, Shao et al. conducted a systematic review for lung cancer and identified four prognostic studies on OS and seven on DFS\PFS. The results of the meta-analysis were not statistically significant for OS: HR 1.26 (95% CI: 0.66–2.40) and DFS\PFS: HR 1.28 (95% CI: 0.82–1.97) [62].

Significant results have been found on the prognostic role of miR-155 in haematopoietic tumors and gliomas. In fact, the results of the meta-analysis conducted by Lu et al. [63], found a significant association between elevated miR-155 expression and poor OS in 2114 patients (HR: 1.72, 95% CI [1.50–1.97]) with haematopoietic tumors, while for Zhouet, gliomas on 1259 patients identified a pooled HR for OS of 1.40 (95% CI [1.19–1.63]) [64].

Other systematic reviews investigated the prognostic role of microRNA-155 in various carcinomas. The findings of He et al. [65] for OS suggest that miR-155 detection has a prognostic value in cancer patients, and that regularly measuring miR-155 expression may be useful in clinical practice, with the pooled HR of 2.057 (95% CI: 1.92–3.039).

The majority of the studies included in the review agreed that the different expression of miR-155 can be a prognostic biomarker for survival for HNSCC patients.

In fact, the HR data on OS for HNSCC in the included studies were all favorable for a worsening of the prognosis in the course of high miR-155 expression, but based on the meta-analysis and an individual analysis of the data, only two studies did not intersect the line of no effect (Figure 2) (Hess et al. and Baba et al.); however, by aggregating the HR data of the six studies, the data became significantly in favor of a worsening of the prognosis (HR = 1.40, CI: [1.13, 1.75] [58]. However, it is low (*I*^2^ 28%), and additionally all measures have been taken to evaluate its source. In fact, it would seem that the sources were the studies of Zhao et al. or Baba et al. [57,58];

The study by Zhao et al. reported an HR of: 1.23 (1.059–1.373) with a standard error of 0.1277, presenting a weight of the data on the meta-analysis of 78% (Figure 2). Nevertheless, even excluding the data of their study, the aggregate HR was always in favor of the worsening of OS (Figure 6). The presence of the heterogeneous data of Zhao et al. could depend on the type of tumor; in this case it was LSCC [57].

The data from Baba et al. in 2016 reported an HR of 5156 for OS; miR-155 was upregulated in both OSCC cell lines and formalin-fixed tissue samples and associated with a poor prognosis [58].

These data were also confirmed in previous studies. In fact, Shi et al. (2015) reported for OSCC an HR of 1.75 for elevated miR-155 expression and added that the elevated level of miR-155 in cancer-free mucosal (ACF) tissues may be related to poor prognosis. Moreover, Shi et al., provided interesting evidence to support the view that the miR-155 was overexpressed in OSCC and located in the cancer nest, the inflammatory area, and the vascular endothelium of OSCC [50].

Regarding the subgroup analysis, our results show the division of the studies in relation to the histological findings, separating the OSCCs from the other HNSCCs. In the OSSC group, the HR (2.72) clearly favored a worsening of OS. However, for the second subgroup, on the one hand, in the forest plot (Figure 7), the black rhombus representation of the final effect touches the central line of the non-effect, while on the other hand, only two studies were reported, those of Hess et al., and Zhao et al. [56,57], limiting the significance of the results.

In addition, the DFS index was reported taking in account three total studies (Baba et al. [58], Kim et al. [59], and Wu et al. [48]). The HR value was in favor of a slight worsening of the DFS index, but the data were not significant. As shown in Figure 3, it is interesting to observe that all the three studies intersect the no-effect line, as well as the measurement of the final effect (the central black diamond). Finally, for the last prognostic index (PFS), the two included studies (Jakob et al. [47] and Bersani et al. [60]) were in contrast, and the resulting HR data (1.09) were due to the absence of both a worsening and/or an ameliorative effect of the PFS (Figure 4) [58].

Furthermore, for the meta-analysis of OS, the TSA was conducted: the results indicated that there was a statistical power in the data, even if from the APIS graphs (Figure 8), with an RRR of 38%; the total ideal number of people with a power of 80% was 475, and for OS the total of the patients included was 470. Even, in the presence of valid statistical power, we can speculate that further studies are needed to support these data.

Although the present meta-analysis was performed following the Cochrane handbook indications, and reported using the PRISMA guidelines, we found some limitations. Data included for DFS and PFS excluding Jakob et al. [47] were extrapolated from the Kaplan Maier curves using the Tierney method, with the aid of dedicated spreadsheets. The method, although validated and used regularly in the meta-analysis in which HR was investigated, was not free from inaccuracies. Furthermore, the data were extrapolated from Kaplan Maier curves with few events (Baba et al. [58] and Kim et al. [59]); therefore, in these conditions, the extrapolated HR value was more prone to errors made by the reviewers [46]. This limit also affected the first meta-analysis performed on the OS, due to the presence of only one study whose data were extrapolated from the curve, a limit which also affected the assessment performed by GRADE.

## 5. Conclusions

In conclusion, even though this meta-analysis is limited, it can be stated that miR-155 could be a promising prognostic biomarker of survival for HNSCC. Consequently, exhaustive investigations of miRNA—for instance, investigations regarding the intercommunication among miRNAs and between miRNAs and other genes, the altered protein expression induced by miRNAs, and site-specific miRNA expression profiling—are prerequisites before future clinical trials of therapeutic applications.

## Figures and Tables

**Figure 1 biology-11-00651-f001:**
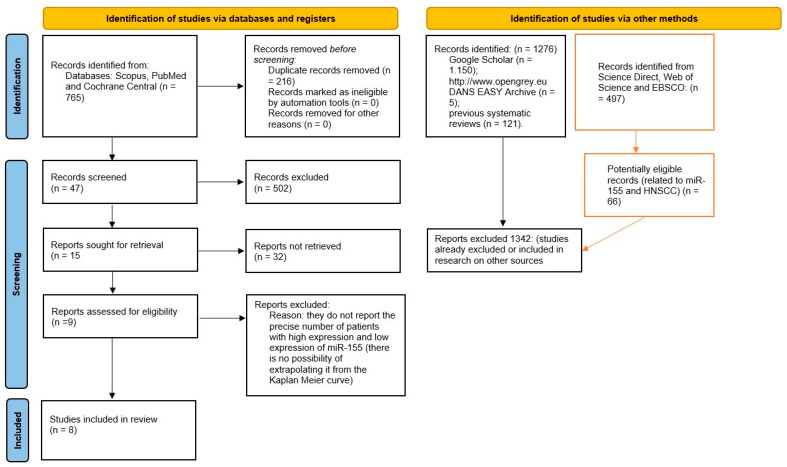
The entire selection and screening procedures are described in the PRISMA; tables with the red lines are the searches performed subsequently (on 12 April 2022), with the addition of Web of Science, Science Direct, and EBSCO.

**Figure 2 biology-11-00651-f002:**
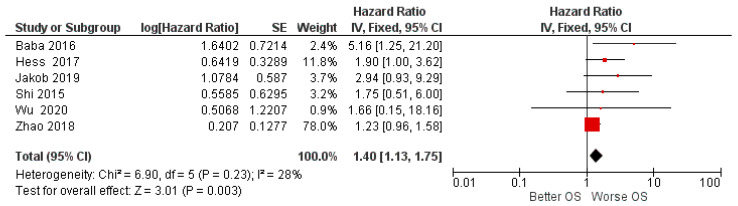
Forest plot of the fixed effects model of the meta-analysis; OS, HR = 1.40, 95% CI: [1.13, 1.75]; df = degrees of freedom; *I^2^* = Higgins heterogeneity index, *I^2^* < 50%, heterogeneity irrelevant; *I^2^* > 75%, significant heterogeneity; C.I. = confidence intervals; *p* = *p* value; SE = standard error. The graph for each study shows, the lead author and the date of publication, the Hazard Ratio with confidence intervals, with the log HR standard error and weight of each study expressed as a percentage. The final value is expressed in bold with the relative confidence intervals. The black line shows the position of the average value, and the rhombus in light black shows the measure of the average effect.

**Figure 3 biology-11-00651-f003:**
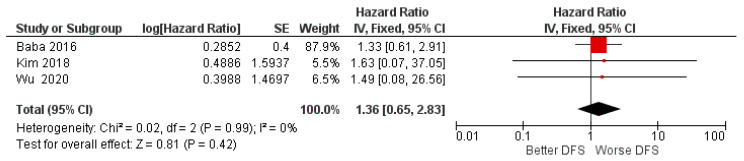
Forest plot of the fixed effects model of the meta-analysis; DFS, HR = 1.36, 95% CI: [0.65 2.83].

**Figure 4 biology-11-00651-f004:**
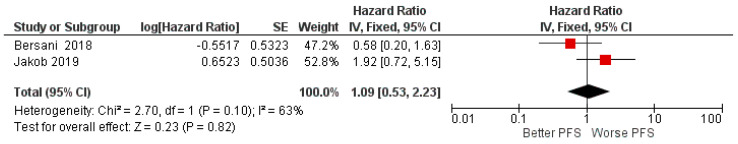
Forest plot of the fixed effects model of the meta-analysis; DFS, HR = 1.09, 95% CI: [0.53 5.15].

**Figure 5 biology-11-00651-f005:**
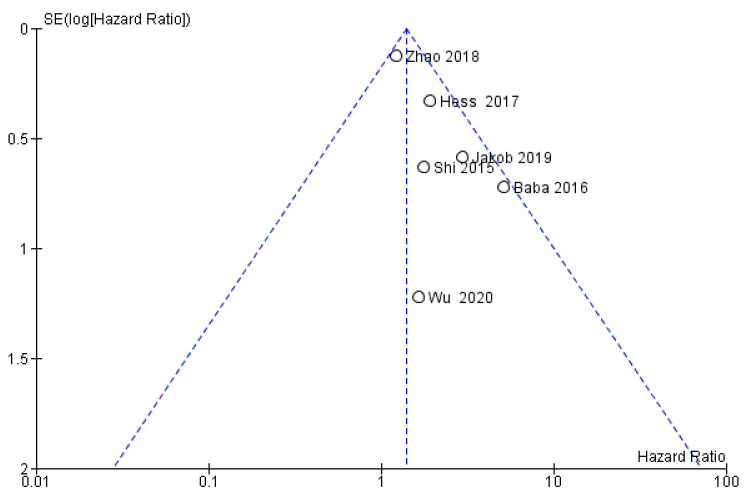
Funnel plot for the first meta-analysis, *I*^2^ = 28%. The absence of heterogeneity is highlighted graphically. SE: standard error.

**Figure 6 biology-11-00651-f006:**
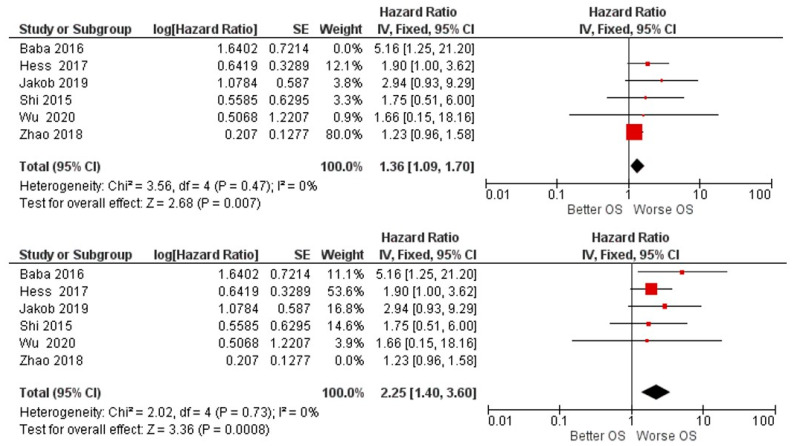
Sensitivity analysis: by excluding the two studies, Zhao 2018 and Baba (2016), the heterogeneity drops (*I*^2^ = 0%), The final effects always remain in favor of a worsening of OS for patients with high expressions of miR-155 tissue. Excluding Baba, HR is 1.36 [1.09, 1.70]; excluding Zhao, HR is 2.25 [1.4, 3.60].

**Figure 7 biology-11-00651-f007:**
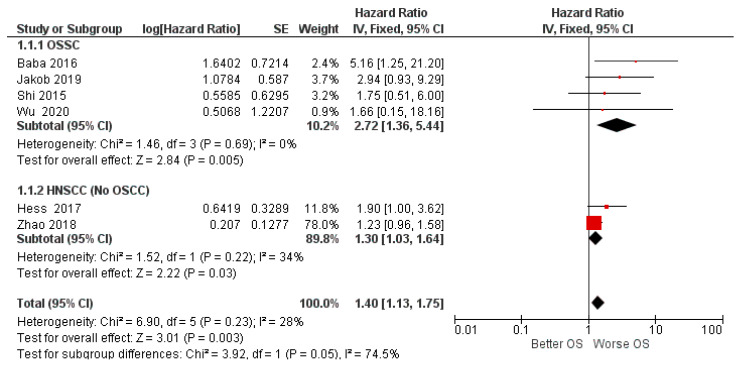
Forest plot of the fixed effects model of the subgroup meta-analysis for OS.

**Figure 8 biology-11-00651-f008:**
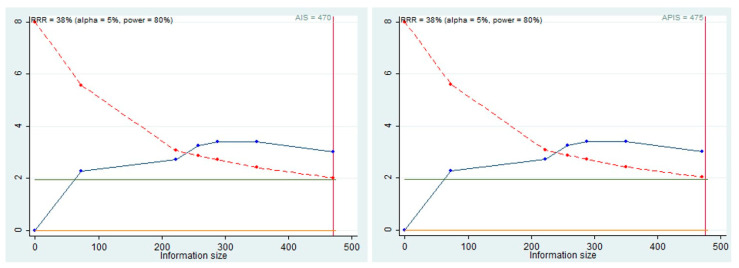
AIS, APIS, light green line (Z = 1.98), dashed red line (monitoring boundary), blue line (cumulative z curve), red line (sample size).

**Table 1 biology-11-00651-t001:** The data extracted for the four articles included in the meta-analysis.

Lead Author, Data	Country	Study Design	Number of Patients (Male, Female); Grading (G1, G2, G3); Staging (I-II, III-IV).	Follow up Max	Tumor Type/Tumor Site	Cut-Off	miR	HR miR-155 Low and High Expression (OS, PFS, CSS, DFS, RFS)
**Jakob (2019) [47]**	Germany	RT	36 (27, 9); G (2,30,4); S (10, 26)	60 months	OSCC	median	miR-21, miR-29, miR-31, miR-99a, miR-99b, miR-100, miR-143, miR-155.	OS:HR 2.94 (0.93–0.29) *p* = 0.005415 RFS:HR 2.04 (0.67–6.2) *p* = 0.19692 PFS: HR 1.92 (0.7–5.22) *p* = 0.19524
**Hess (2017) [56]**	Germany	RT	149 (123, 26)	61 months	HNSCC (oropharynx, *n* = 78; hypopharynx, *n* = 71)	median	miR-155, miR-200b, miR-146a.	OS: HR 1.90 (1–1.37) *p* = 0.051
**Zhao (2018) [57]**	China	RT	120 (107, 13);	79 months	LSCC	median	miR-155.	OS: HR 1.23 (1.059–1.373) *p* = 0.105
**Baba (2016) [58]**	Japan	RT	73(49, 24); S (29, 44)	24 months	OSCC	median	miR-155.	OS: HR 5.156 *p* = 0.023 DFS:HR 1.3300 *p* = 0.47 ^1^
**Shi (2015) [50]**	China	RT	30 (19, 11); S (8, 22).	31.93 months (5–50)	OSCC	median	miR-155.	OS: HR 1.748 (0.508–6.015) *p* = 0.375
**Kim (2018) [59]**	Korea	RT	68 (45, 23); S (15, 19).	80 months	OSCC	median	miR-155.	DFS:HR 1.6300 *p* = 0.7592 ^1^
**Bersani 2018 [60]**	Sweden	RT	168 (126, 42); S (17, 151).	34 months	TSCC/BOTSCC ^2^	Quartile	miR-155, miR-185, miR-193b.	PFS: HR 0.5760 *p* = 0.30 ^1^
**Wu (2020) [48]**	China	RT	62 (42, 20); S (46, 16).	60 months	OSCC	median	miR-155.	OS:HR 1.6600 *p* = 0.6780 ^1^ DFS:HR 1.4900 *p* = 0.7861 ^1^

^1^ data extracted from Kaplan Meier survival curves, ^2^ tonsillar squamous cell carcinoma (TSCC), base of tongue squamous cell carcinoma (BOTSCC), S (Staging), G (Grading).

**Table 2 biology-11-00651-t002:** Assessment of risk of bias within the studies. with scores 8 to 10 = low quality, 11 to 14 = intermediate quality, and 15 to 18 = high quality.

Lead Author, Data	Sample	Clinical Data	Marker Quantification	Prognostication	Statistics	Classical Prognostic Factors	Score
**Jakob (2019) [47]**	1	3	3	3	3	3	16
**Hess (2017) [56]**	3	2	3	2	2	2	14
**Zhao (2018) [57]**	3	2	3	2	2	2	14
**Baba (2016) [58]**	2	3	3	2	2	2	14
**Shi (2015) [50]**	1	2	3	2	2	2	12
**Kim (2018) [59]**	2	2	3	2	2	1	12
**Bersani (2018) [60]**	3	3	3	2	2	1	14
**Wu (2020) [48]**	2	2	3	3	3	2	15

**Table 3 biology-11-00651-t003:** Evaluation of GRADEpro GDT.

Certainty Assessment							№ of Patients	Effect		Certainty
№ of Studies	Study Design	Risk of Bias	Inconsistency	Indirectness	Imprecision	Other Considerations		Relative (95% CI	Absolute (95% CI)	
**6**	observational studies	not serious	not serious	not serious	Serious ^1^	all plausible residual confounding would suggest spurious effect, while no effect was observed	470	HR 1.40 (1.13 to 1.75)	2 fewer per 1.000 (From 2 fewer to 1 fewer)	⨁⨁◯◯ Low

^1^ In Wu et al. (2020), the HR value was extrapolated from the Kaplan Meier curves reported in the manuscripts; this passage is not free from error and can be a source of inaccuracy. CI, Confidence Interval; HR, Hazard Ratio.

## Data Availability

Not applicable.

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
