# Peer review of "Biological Prognostic Value of miR-155 for Survival Outcome in Head and Neck Squamous Cell Carcinomas: Systematic Review, Meta-Analysis and Trial Sequential Analysis"

_biology, 2022, doi:10.3390/biology11050651_

Round 1

Reviewer 1 Report

Dioguardi et al. explored the clinical utility of tissue miR-155 as potential prognostic biomarker in head and neck squamous cell carcinoma. Although this is an interesting topic, some issues should be remained.

1-In Introduction Section the authors should include relevant references that support the content of this section. Lines 59-61 (please include references), lines 66-71 (include relevant references in this field like PMID: 31756680, PMID: 34646767…..).

2-The authors did not perform the literature search in other important databases such as Web of Science, Embase and Science Direct.

3-I would have evaluated the quality of the studies with Quality assessment tool developed by the National Heart, Lung and Blood Institute (NHLBI) (USA) for Observational and Cross-sectional studies. Conclusions must be based with on sound and good research methodology.

5- Was the potential of publication bias evaluated?

6-The authors should include in Material and Methods Section a paragraph about subgroup analysis (PFS or DFS) and heterogeneity analysis.

7-The authors did not discuss their results with previous investigations based on the prognostic potential of microRNAs in head and neck cancer (PMID= 31349668).

8-What is the prognostic value of the miR-155 in another tumors? The potential of this biomarker should be discussed as a cancer biomarker.

6- Abbreviations are not uniform along the manuscript. Please check all abbreviations. Line 82 (miRNA) vs. Line 85 (Mir) vs Line 98 (mIr) vs Line 147 (Microrna) vs Line 265 (mirR), Line 348 (hazard ratio)………..

7-There are several mistakes. Line 233 (I2 28%), Line 236 (Higgs index), Line 134 (“k words”), Line 369 (show)……

Author Response

Dioguardi et al. explored the clinical utility of tissue miR-155 as potential prognostic biomarker in head and neck squamous cell carcinoma. Although this is an interesting topic, some issues should be remained.

  • 1-In Introduction Section the authors should include relevant references that support the content of this section. Lines 59-61 (please include references), lines 66-71 (include relevant references in this field like PMID: 31756680, PMID: 34646767…..).
  • 2-The authors did not perform the literature search in other important databases such as Web of Science, Embase and Science Direct.
  • 3-I would have evaluated the quality of the studies with Quality assessment tool developed by the National Heart, Lung and Blood Institute (NHLBI) (USA) for Observational and Cross-sectional studies. Conclusions must be based with on sound and good research methodology.
  • 5- Was the potential of publication bias evaluated?
  • 6-The authors should include in Material and Methods Section a paragraph about subgroup analysis (PFS or DFS) and heterogeneity analysis.
  • 7-The authors did not discuss their results with previous investigations based on the prognostic potential of microRNAs in head and neck cancer (PMID= 31349668).
  • 8-What is the prognostic value of the miR-155 in another tumors? The potential of this biomarker should be discussed as a cancer biomarker.
  • 6- Abbreviations are not uniform along the manuscript. Please check all abbreviations. Line 82 (miRNA) vs. Line 85 (Mir) vs Line 98 (mIr) vs Line 147 (Microrna) vs Line 265 (mirR), Line 348 (hazard ratio)………..
  • 7-There are several mistakes. Line 233 (I28%), Line 236 (Higgs index), Line 134 (“k words”), Line 369 (show)……

ANSWER

Thank you for your suggestions and advice. Your comments have been enormously helpful in improving the manuscript especially in the risk of bias part (which has been implemented with the addition of an additional tool) and publication bias. Furthermore, the addition of 3 further databases as suggested to me served to minimize the publication bias which represents one of the problems that science has to face (the material not published because it does not report statistically significant data).

  • Bibliographic references supporting the contents in paragraphs 59 to 71 have been added as required: At the base of the etiopathogenetic processes of cancerization we have the genetic alterations that involve mutations of the main oncogenes (Ras [1], raf [2], gsp [3], jun [4], fas [5], erbA [6], abl [7], sis [8], erbB [9] and fms [10]) and oncosuppressor (APC [11], p53 [12], NF1 [13], VHL [14], Rb [15], BCL2 [16], BRCA2 [17], PTCH [18], CD95 [19], ST5 [20], SWI / SNF [21], p16 [22] , YPEL3 [23], ST7 [24] and ST14 [25]). Understanding of the genomic aberrations linked with noncoding genes (such as microRNA s) and their effects on HNSCC is still relatively limited. Current evidence suggests that deregulation of microRNA genes plays an important role in HNSCC [26-36].

MicroRNAs (miRNA, miR), are a large group of small single-stranded non-coding endogenous RNAs approximately 18-25 nucleotides in length that play a significant role in the post-transcriptional regulation of genes through the interaction with 3'UTR of target mRNA [37]. The degradation or inhibition of its translation, also play an important role for cell development in differentiation, metabolism, proliferation, migration, induction of angiogenesis and apoptosis, and are stable molecules that can be found not only in tissues but also in body fluids such as blood, saliva and urine [38].

  • 3 more databases have been added such as Web of science, Science direct and EBSCO
  • a risk assessment or f bias was also performed through the Quality assessment tool developed by the National Heart, Lung and Blood Institute (NHLBI) (USA) for Observational and Cross-sectional studies, the table was added in the supplementary materials and was cited in the text.
  • An evaluation of the publication bias was also carried out as suggested, the evaluation was carried out through the graphical analysis of the symmetry of the funnel plot, which appears asymmetrical. The presence of asymmetry demonstrates the potential for publication bias. in order to minimize bias, a search for unpublished material in the gray literature was carried out which also includes the presence of lecture proceedings and material that may not have been published due to the lack of significance of the results.

  • added the following paragraph in the materials and methods section as required: The evaluation of the heterogeneity will be performed through the Higgins index (I2) and the Chi2, values of I2 higher than 75% will lead to a moderate heterogeneity of the data in the studies. The heterogeneity will also be evaluated graphically through the analysis of the overlapping of the confidence intervals in the forest plot, through the graphical analysis of the funnel plot to search for any sources of heterogeneity on the presence of a publication bias.

 A subgroup analysis will also be performed with different meta-analyzes as a function of the various prognostic indices of survival and as a function of the histological sub-types of HNSCC.

  • Added as required in discussion the following paragraph:Only a previous systematic review of the literature on the prognostic value of several mirs, including miR-155 in head and neck cancers, included miR-155 in a sub-group analysis that included only 2 studies (Shi et al. [48], Hess et. al. [54]). The results of this meta-analysis, HR of 1.866 (95% CI 1.047–3.326), are in line with our systematic review, which identified as many as 6 studies involving HR on OS [59].
  • Added as required in discussion the following paragraph: Prognostic value of the altered expression was also investigated for other tumors in fact Shao et al conducted a systematic review for Lung cancer and identified 4 prognostic studies on OS and 7 on DFS \ PFS, the results of the meta-analysis were not statistically significant OS: HR 1.26 (95% CI: 0.66–2.40) and DFS \ PFS: HR 1.28 (95% CI: 0.82–1.97) [60].

Significant results are had on the prognostic role of miR-155 in haemopietic tumors and glioma in fact the results of the meta-analysis conducted by Lu et al. He found a significant association between elevated miR-155 expression and poor OS in 2114 patients (HR: 1.72, 95% CI [1.50-1.97]) with haematopoietic tumors [61] while for Zhouet glioma on 1259 patients identified a pooled HR for OS of 1.40 (95% CI [1.19-1.63]) [62]. 

Other systematic reviews, however, investigated the prognostic role of aggregated tissue miR-155 the results of studies on different tumors: He et al for overall survival, higher miR-155 expression could significantly predict worse outcome with the pooled HR of 2.057 (95% CI : 1,392-3,039)[63]

  • all abbreviations have been corrected
  • all reported errors have been corrected

Reviewer 2 Report

The manuscript entitled:"Biological Prognostic Value of miR-155 for Survival Outcome in Head and Neck Squamous Cell Carcinomas: Systematic Re
view, Meta-Analysis and Trial Sequential Analysis" focused on a systemic revision of literature data about the role of mir-155 in the clinical section of head and neck squamous cell carcinoma patients is well written but requires moderate revisions to be accepted for the publication

  • In the introduction section, please, could the authors better define the role of miRNA in the clinical stratification of head and neck tumor patients?
  • In the introduction sectio (liens 60- 62), please could the authors better report genes? In details, i woudl suggest to define the extensive name for each with capital letter.
  • In the results section, please, could the authros also focus on the evaluation of clinical and morphological features (such as, sex, gender,  grading, staging) as statistically relevant factors that could impact on survial?
  • In my opinion table 1 may be transferred in supplementary file,; in this section, the description of search modality is adequate
  • Please, could the authors review figure style? They are not in line with the features of text.
  • In the research strategy, please, could the authors ebvaluate if other rleevant paper may be included in this metaanalysis? In my opinion, this aspect should eb reviewed

Author Response

The manuscript entitled:"Biological Prognostic Value of miR-155 for Survival Outcome in Head and Neck Squamous Cell Carcinomas: Systematic Re
view, Meta-Analysis and Trial Sequential Analysis" focused on a systemic revision of literature data about the role of mir-155 in the clinical section of head and neck squamous cell carcinoma patients is well written but requires moderate revisions to be accepted for the publication

  • In the introduction section, please, could the authors better define the role of miRNA in the clinical stratification of head and neck tumor patients?
  • In the introduction sectio (liens 60- 62), please could the authors better report genes? In details, i woudl suggest to define the extensive name for each with capital letter.
  • In the results section, please, could the authros also focus on the evaluation of clinical and morphological features (such as, sex, gender, grading, staging) as statistically relevant factors that could impact on survial?
  • In my opinion table 1 may be transferred in supplementary file, in this section, the description of search modality is adequate
  • Please, could the authors review figure style? They are not in line with the features of text.
  • In the research strategy, please, could the authors ebvaluate if other rleevant paper may be included in this metaanalysis? In my opinion, this aspect should eb reviewed

ANSWER

Thank you for the comments and suggestions that have been enormously helpful in improving the manuscript especially in the aspects of clinical stratification of patients with HNSCC and the role of miR alteration as an expression of other RISK factors such as papilloma virus and Smoking.

  • As requested, I have added more information in the introductory section concerning the role of mirRs in the clinical stratification of HNSCC patients:

There is a lot of evidence linking the altered profile of miRs with well-defined disease states affecting the onset, progression and metastasis of HNSCC. Their possible use as potential prognostic biomarkers can help to identify potential clinical characteristics of malignancy such as relapses, invasiveness and metastases in order to be able to modulate a more effective surgical, chemotherapy and radiotherapy treatment in relation to the prognosis of the patient. There also seems to be an association that links patients with HNSCC HPV + with an altered expression of miRs, in fact in a study conducted by Emmett et al. in 2021 [1] who compared miRs, stratified by HPV status in HNSCC showed the presence of 5 miRs (miR-16-3p, miR-29a, miR-29c, miR-150 and miR-363) with different expression. On the other hand, considering smoking patients, a very recent study on HNSCC has identified NNK (tobacco-specific nitrosamine 4- (methylnitrosamino) -1- (3-pyridyl) -1-butanone) present in cigarette smoke as a significant induced factor upregulation of miR-21 and miR-155 and downregulation of miR-422 [2]

  • modified as required the names of the genes
  • in the results section, the gender aspects of staging and grading have been extended as required, furthermore the relative data extracted from the studies have been included in table 1.:Only one study reported information on the histological grading of HNSCC, while the stratification of patients according to staging was clearly present in 6 of 8 studies. The total of stage I and II patients was 125 out of 403 while those in stage III and IV were 276. In relation to gender, male patients were 538 while female patients were 171, in line with the most recent epidemiological studies.
  • Transferred as required in supplementary files
  • Figure 1 has been redone, the other figures present in the text have been checked.

Tthe search for new records was updated with the inclusion of 3 new databases EBSCO, WEB OF SCIENCE and SCIENCE DIRECT in addition to PUBMED, SCOPUS and Cohcrane libray, in addition, the gray literature was consulted through the DANS register (openGRAY) as well as google scholar for the identification of additional unpublished material in order to minimize publication bias.

  1. Emmett, S.E.; Stark, M.S.; Pandeya, N.; Panizza, B.; Whiteman, D.C.; Antonsson, A. MicroRNA expression is associated with human papillomavirus status and prognosis in mucosal head and neck squamous cell carcinomas. Oral Oncol 2021, 113, 105136, doi:10.1016/j.oraloncology.2020.105136.
  2. Doukas, S.G.; Vageli, D.P.; Lazopoulos, G.; Spandidos, D.A.; Sasaki, C.T.; Tsatsakis, A. The Effect of NNK, A Tobacco Smoke Carcinogen, on the miRNA and Mismatch DNA Repair Expression Profiles in Lung and Head and Neck Squamous Cancer Cells. Cells 2020, 9, doi:10.3390/cells9041031.

Round 2

Reviewer 1 Report

The authors have performed corrections following the indications. However, should issues should be addressed: 

-Line 428-451. This part of the discussion section it is difficult to understand for the reader. Authors should improve the written content. Besides, there are several mistakes: "Lung cancer", "overall survival", "haemopietic"....

Author Response

We thank the Reviewers for the considerable attention and the valuable comments that certainly helped us to improve the quality of the present paper. We have revised the manuscript according to the Reviewers’ comments. A revision of the article has been carried out (underlined in green in the main manuscript text).

Please let us know if the revised paper satisfies requirements for publication.

Thank you very much for your attention and courtesy.

We have really appreciated the important suggestions of the reviewer. The unclear sentences, as well the typos errors, were deeply revised, according to referee's kind request. Thank you very much.

Reviewer 2 Report

The manuscript may be accepted in the present form

Author Response

(The authors gave the same response as above.)
